# Evaluation of 10-Week Neuromuscular Training Program on Body Composition of Elite Female Soccer Players

**DOI:** 10.3390/biology11071062

**Published:** 2022-07-17

**Authors:** Alberto Roso-Moliner, Elena Mainer-Pardos, José Luis Arjol-Serrano, Antonio Cartón-Llorente, Hadi Nobari, Demetrio Lozano

**Affiliations:** 1Health Sciences Faculty, Universidad San Jorge, Autov A23 km 299, Villanueva de Gállego, 50830 Zaragoza, Spain; aroso@usj.es (A.R.-M.); jlarjol@usj.es (J.L.A.-S.); acarton@usj.es (A.C.-L.); dlozano@usj.es (D.L.); 2Faculty of Sport Sciences, University of Extremadura, 10003 Cáceres, Spain; hadi.nobari1@gmail.com or; 3Department of Motor Performance, Faculty of Physical Education and Mountain Sports, Transilvania University of Braşov, 500068 Braşov, Romania; 4Sports Scientist, Sepahan Football Club, Isfahan 81887-78473, Iran; 5Department of Exercise Physiology, Faculty of Educational Sciences and Psychology, University of Mohaghegh Ardabili, Ardabil 56199-11367, Iran

**Keywords:** football, body fat, women, strength training, lean body mass, kinanthropometry

## Abstract

**Simple Summary:**

Soccer performance is complex, requiring mastery of sport specific technical and tactical skills with ideal physical fitness (i.e., includes sprints, hops, accelerations, changes of directions, and so on) and body composition (i.e., increase lean muscle mass and decrease fat mass). In the last decades, performance models have helped to understand the multifactorial mechanisms involved in physical performance in sports. Hence, we tested the hypothesis that a neuromuscular training (NMT) program has an effect on body composition parameters in elite female soccer players. The result showed that implementation of 10-week with thrice-weekly NMT program improves body composition in elite female soccer players.

**Abstract:**

(1) Background: This study was conducted to investigate the effects of a 10-week neuromuscular training program (NMT) on the sum of six skinfolds (Σ6S) and body composition variables in elite female soccer players. (2) Methods: Forty-four Spanish elite female soccer players (age: 24.0 ± 4.2 years; height: 164.3 ± 5.5 cm; body mass: 60.4 ± 5.5 kg; body mass index (BMI): 22.4 ± 2.2 kg/m^2^) were randomly assigned to a control group (CG) or to an experimental group (EG). Participants in the EG completed a specific NMT program of 24 min, three times per week, which included exercises from six different categories (mobility, dynamic stability, anterior chain strength, lumbopelvic control, posterior chain strength, and change of direction). The CG followed their normal strength and conditioning program. Pre- and post-intervention assessments included anthropometric measurements (weight, height, limb circumferences, and bone breadths), and subsequently, body composition factors BMI, Σ6S, body mass, muscle mass, and lean body mass were calculated. Nutrition was standardized by a nutritionist and also load monitored. (3) Results: A two-way mixed analysis of variance (group × time) revealed that there was a significant (*p* ≤ 0.001) group × time interaction between body mass, fat mass, and Σ6S in favor of NMT. A significant interaction was also observed for body skeletal muscle mass and lean body mass favoring NMT. (4) Conclusions: The application of an NMT program seems to be a useful strategy to improve body composition in elite female soccer players.

## 1. Introduction

Soccer is arguably the most popular team sport today, and it is played by more than one billion people worldwide [1]. In 2020, the number of federated female soccer players in Spain reached 77,461 [2]. That year, FIFA’s Women’s Football Strategy set out its goals of doubling the number of participants by 2026 [3].

Soccer performance is multifactorial and requires mastery of both sport-specific technical–tactical skills and optimal physical fitness. As an intermittent high-intensity sport, soccer involves activities such as sprints, jumps, accelerations, and changes of direction (COD), among others [4,5]. Of note, these high-intensity actions, coupled with the ability to repeat them without fatigue and the somatotype, also account for most actions that cause injury [6]. In addition, the somatotype of the players has shown to be relevant for this purpose [7]. Therefore, the link between anthropometry, muscle performance, and soccer-specific physical performance has been studied extensively (i.e., sprint, repeated sprint ability, vertical jump, etc.). Previous investigations found a strong link between body composition (high levels of lean mass and low levels of fat mass) and vertical jump performance and repeated sprint ability [8,9,10]. In addition, for leaner body compositions, lower body strength measurements are closely linked to soccer players’ acceleration, sprinting, and leaping performance [11,12]. Given these correlations, training methods, such as neuromuscular training (NMT), that increase lower body strength and/or reduce body fat, enhancing the power-to-mass ratio, should result in significant increases in the physical performance parameters of female soccer players [13,14].

In this direction, the relationship between running performance (i.e., aerobic capacity) and body composition has been evaluated in elite males [15], youths [16], and elite females [17], showing high-speed actions and longer distances covered in players with greater lean body mass percentages as a marker of the muscle-to-fat ratio. In addition, previous published kinanthropometric studies in soccer showed different profiles as a function of age, sex, and playing position, and some of them have specifically analyzed the anthropometric profile of female soccer players [18,19,20,21].

The research working group on the body composition health and performance of athletes states that low body fat and high lean body mass are strongly correlated with higher levels of performance, especially in weight-sensitive sports such as soccer [22]. However, this relationship should be handled with caution because each sport has its own body composition (i.e., somatotype) that is considered ideal for success [23], and players with low body fat mass do not follow this general rule. A recently published review included kinanthropometric data of elite female soccer players from 2000 to 2020, showing a fat mass percentage between 14.5% to 22% [24]. In addition to the above, it is also important to analyze other performance factors that affect this sport, such as adequate nutritional intake [25] and genetics [26]. A few studies looked at the relationship between different endocrine parameters, such as IGFBP-3, erythropoietin, or estrogen, in female athletes. Additionally, genes related to performance and body composition, such as angiotensin-1 converting enzyme insertion/deletion (ACE I/D) polymorphism or α-actinin-3 (ACTN3) R577X polymorphism, have been studied [11].

In the last two decades, novel training approaches have been developed aiming to improve performance and body composition in female soccer players [17,18,19,22,23,24,27,28]. Interestingly, Myer et al. [29] suggested an integrative NMT program including mobility, dynamic stability, core strength, plyometric, agility, and fundamental strength exercises, showing it could improve sport-specific skills and minimize the risk of injuries.

With this rationale, some standardized neuromuscular protocols, such as FIFA11+, Sportsmetrics^TM^, or Harmoknee, were developed and demonstrated to reduce injury risk [30,31,32,33,34,35] and improve performance [31,36] in female soccer players. Despite the effects of these training protocols on performance, the effect on body composition remains unknown. Rohmansyah et al. [37] found a reduction in body mass index (BMI), fat mass, and waist circumference in obese young-adult females after a 6-week FIFA 11+ program. Simões et al. analyzed the effects of NMT on body composition in volleyball athletes, finding improvements in body composition [38]. With respect to each of the interventions included in an NMT program, there is some research on the effects of body-weight resistance training [39], eccentric [40], and plyometric-based programs [41] on the body composition of female soccer players. However, the evidence is very scarce regarding the effects of a multicomponent program that combines all of them. Due to this, in the present study we hypothesized that an NMT program can be used to increase the lean body mass and body skeletal muscle mass or reduce the body mass, BMI, fat mass, and skinfold measurements in female soccer players. Thus, the main aim of this study was to evaluate the effects of an NMT program on the body composition of elite female soccer players.

## 2. Materials and Methods

### 2.1. Participants

Forty-four Spanish, highly-trained female soccer players voluntarily participated in the study (Table 1).

Data collection took place during the competitive period (i.e., seventh month of the season). All the participants played for soccer teams in the Spanish Women’s Second Division and completed a similar weekly soccer training regarding volume and methodology (i.e., five 90 min sessions per week and 1 game/week). All the participants met the following inclusion criteria: (i) at least 6 years of experience in soccer training and competition; (ii) participation in regular soccer training and competition for 6 months before data collection; (iii) free from injuries, and iv) refrained from other NMT programs or diets outside this study. Furthermore, participants were excluded if: (i) they missed three or more NMT sessions or (ii) they missed a testing day. The participants were randomly assigned (ABBA distribution) to a control group (CG, *n* = 22) or to an experimental group (EG, *n* = 22). Nevertheless, due to NMT attendance and testing days, the final sample was *n* = 22 for CG and *n* = 18 for EG (Figure 1). Prior to data collection, written informed consent was obtained from all the participants. The study was developed following the ethical standards of the Declaration of Helsinki and was approved by the Local Ethics Committee of Clinical Research (PI21/011, CEICA, Spain).

### 2.2. Measurement of Body Composition

The participants were tested according to the guidelines of the International Society of the Advancement of Kinanthropometry (ISAK) at the beginning (i.e., 1 week before) and the end of the intervention (i.e., 1 week after). The instruments were adjusted before their use and data were collected in duplicate [42]. To minimize measurement variation, the same experienced researcher examined all the subjects on the right side of the body during the same time of the day (i.e., 08:00 a.m.–10:00 a.m.). Participants were asked to avoid vigorous activities for at least 48 h before data collection and consumption of large volumes of water 2 h before as well as to follow their ordinary diet. Furthermore, to avoid any possible dietary confounding effects on body-composition assessment, in the pre- and post-test sessions, a 24-h food recall was collected by a registered dietician to check average macronutrient and energy intake (DAPA Measurement Toolkit, Cambridge, UK). After obtaining the players’ data, the Spanish Food Composition Database (BEDCA) was used to calculate kilocalories and macronutrient intake. This database includes a compilation of nutritional data from various publications and food composition tables [43]. The results are shown in Table 2.

Anthropometric measurements included: body mass in kilograms (kg) using a digital scale (BC-601, Tanita, IL, USA), height in centimeters (cm) employing a stadiometer (SECA 214, SECA, Hamburg, Germany), limb girths in cm using an anthropometric tape (Lufkin W606PM, Lufkin, NC, USA), bone breadths in cm utilizing a bone caliper (Campbell 10, Rosscraft, CA, USA), and skinfolds in millimeters (mm) using a slim guide skinfold caliper (Harpenden, West Sussex, UK). Specifically, 8 point skinfolds (e.g., triceps, biceps, abdominal, iliac crest, supraspinal, subscapular, front thigh, and medial calf), 4 limb girths (e.g., arm relaxed, arm tensed, mid-thigh, and calf), and 3 bone breadths (e.g., biepicondylar humerus, biepicondylar femur, and bi-styloid diameter of the wrist) were measured. The inter- and intra-observer technical error of measurement was less than 5.5% for skinfolds and less than 1.5% for the other variables.

BMI was calculated as body mass (in kg) divided by height in meters squared (kg/m^2^) [44]. The sum of six skinfolds (Σ6S) was obtained as the addition in mm of the standardized 6 skinfolds (triceps, subscapular, supraspinal, abdominal, front thigh, and medial calf) [21]. Body density (BD) was calculated using the equation proposed by Withers et al. (1987) for female athletes [45] (Equation (1)). When BD was calculated, the Siri equation [46] was used to estimate fat mass percentage (Equation (2)). Lean body mass percentage was calculated as the difference between total body mass percentage and fat mass percentage. The body skeletal muscle mass was estimated with the equation of Lee et al. (2000) [47], and once this result was obtained, we converted it to a percentage (Equation (3)).
(1)1.17484 – {0.07229 ∗ [Log (Σ4S.triceps+subscapular+supraspinal+medial calf)]}
(2)[(4.95/BD) − 4.5) ∗ 100]
(3)Ht ∗ (0.00744 * CAG2+0.00088 ∗ CTG2+0.00441 ∗ CCG2)+(2.4 ∗ sex ) − (0.048 ∗ age )+race+7.8

Ht: height (m); CAG: corrected arm girth (cm); CTG: corrected thigh girth (cm); CCG: corrected calf girth (cm); Sex (1 for male and 0 for female); Race (−2 for Asian, 1.1 for African American and 0 for white or Hispanic)

### 2.3. Exercise Protocol

Participants in the EG completed a rise, activate, mobilize, and potentiate (RAMP) system warm-up protocol [48] followed by a 24 min NMT program, three times per week, for 10 weeks. Participants in the CG followed the same warm-up protocol. Then, they completed their normal strength and conditioning program for 24 min (Table 3). Both groups were exercised equally and there was no significant difference between groups. Two weeks before the beginning of the intervention, four familiarization sessions were executed in the EG to get to know the exercises included in the NMT program (Figure 2).

The intensity of the CG and EG training sessions was recorded using the modified Borg scale (0–10 rating), which is valid to control the training intensity and is commonly used by these players in their physical preparation [49,50]. At the end of the physical preparation, the players individually indicated their level of perceived exertion. The average for each of the sessions is shown in Figure 3.

The NMT program (Figure 4) included exercises from six different categories: (1) mobility, (2) dynamic stability, (3) anterior chain strength, (4) lumbopelvic control, (5) posterior chain strength, and (6) COD and was carried out in 4 sets of the 6-exercise circuit (40 s of work and 20 s of gentle running to change to the next exercise). Level 1 exercises were performed during the first 2 weeks, whereas levels 2 and 3 were performed during weeks 3–6 and 7–10, respectively. For unilateral exercises, the working leg changed between series.

### 2.4. Statistical Analysis

Data analysis of the present study was carried out as both descriptive and inferential. Normality was inspected for all variables using a Shapiro-Wilk test. Macronutrient and energy intake, descriptive data, and possible differences pre-training were analyzed with independent group *t*-student. Within-group comparisons (Student paired *t*-test) were carried out to detect significant differences between the pre-test and post-test in all variables in both groups. A 2 (group) × 2 (time) repeated measures ANOVA with Bonferroni post hoc analysis was calculated for each parameter. Hedges’ g effect size with a 95% confidence interval was also calculated to determine the magnitude of pairwise comparisons for pre- and post-test and was defined as trivial (<0.2), small (>0.2), moderate (>0.5), and large (>0.8). If the results of the independent sample *t*-test and effect sizes were similar for each group, then the percentage changes were computed and assessed. The significance of statistical analysis was used at the level of *p* < 0.05. All statistical calculations were performed using SPSS (Version 28.0, IBM SPSS Inc., Chicago, IL, USA).

## 3. Results

The descriptive characteristics of the players of both groups are shown in Table 1. The results of the analysis showed that there were no significant differences between the two groups in these variables. The average intakes of macronutrients and energy are shown in the Table 2. The results of the analysis showed that there were no significant differences in diet during the intervention in either group.

Table 4 shows the mean and SD of the changes in skinfold variables. At the baseline, there were no differences observed between groups in the above variables, except subscapular skinfold (f = 4.91; *p* = 0.033) and sum of six skinfolds (Σ6S) (f = 4.43; *p* = 0.04).

There were significant main effects of time (*p* ≤ 0.001, f = 24.52, η_p_^2^ = 0.39; *p* ≤ 0.001, f = 25.46, η_p_^2^ = 0.40; *p* ≤ 0.001, f = 19.81, η_p_^2^ = 0.34; *p* = 0.009, f = 7.49, η_p_^2^ = 0.16; *p* = 0.007, f = 8.00, η_p_^2^ = 0.17; *p* ≤ 0.001, f = 24.98, η_p_^2^ = 0.39) and a group by time interaction (*p* ≤ 0.001, f = 29.73, η_p_^2^ = 0.44; *p* ≤ 0.001, f = 47.25, η_p_^2^ = 0.55; *p* ≤ 0.001, f = 41.72, η_p_^2^ = 0.52; *p* ≤ 0.001, f = 51.08, η_p_^2^ = 0.57; *p* ≤ 0.001, f = 25.22, η_p_^2^ = 0.41; *p* ≤ 0.001, f = 68.87, η_p_^2^ = 0.64) for front thigh, medial calf, subscapular, iliac crest, abdominal, and Σ6S, respectively. The post hoc analysis indicated that front thigh (EG, *p* ≤ 0.01, *g* = −0.09), medial calf (EG, *p* ≤ 0.001, f = 24.99, η_p_^2^ = 0.40), subscapular (EG, *p* ≤ 0.01, *g* = −0.12), iliac crest (CG, *p* = 0.02, *g* = 0.02, EG, *p* ≤ 0.01, *g* = −0.04), abdominal (EG, *p* ≤ 0.01, *g* = −0.06) and Σ6S (CG, *p* = 0.019, *g* = 0.01 and EG, *p* ≤ 0.01, *g* = −0.09) skinfolds were significantly reduced. Percent changes of skinfold variables between pre- and post-test are shown in Table 4 and Figure 5.

Table 5 shows the mean and standard deviation in body mass, BMI, fat mass, body skeletal muscle mass, and lean body mass. At the baseline, there were no differences observed between groups in the above variables, except the body skeletal muscle mass Lee (f = 16.71; *p* ≤ 0.001). There were significant (*p* = 0.071, f = 8.17, η_p_^2^ = 0.18; *p* = 0.006, f = 8.50, η_p_^2^ = 0.18; *p* ≤ 0.001, f = 16.39, η_p_^2^ = 0.30; *p* ≤ 0.001, f = 32.85, η_p_^2^ = 0.46; *p* ≤ 0.001, f = 16.39, η_p_^2^ = 0.30) main effects of time and a group by time interaction (*p* ≤ 0.001, f = 14.77, η_p_^2^ = 0.28; *p* ≤ 0.001, f = 14.72, η_p_^2^ = 0.28; *p* ≤ 0.001, f = 50.19, η_p_^2^ = 0.57; *p* ≤ 0.001, f = 50.61, η_p_^2^ = 0.57; *p* ≤ 0.001, f = 50.19, η_p_^2^ = 0.56) for the body mass, BMI, fat mass Withers, body skeletal muscle Lee, and lean body mass, respectively. Post hoc analysis found that the body mass (EG, *p* ≤ 0.001, *g* = −0.04), the BMI (EG, *p* ≤ 0.001, *g* = −0.04), fat mass Withers (CG, *p* = 0.029, *g* = 0.02, EG, *p* ≤ 0.01, *g* = −0.10), the body skeletal muscle mass Lee (EG, *p* ≤ 0.001, *g* = 0.23), and lean body mass (CG, *p* = 0.03, *g* = −0.02, EG, *p* ≤ 0.001, *g* = 0.45) were significantly reduced post-test vs. pre-test. Percent changes of all body composition variables between pre- and post-test, as shown in Table 5 and Figure 6 and Figure 7.

The average intensity registered using the modified Borg’s scale (0–10) was recorded over the 30 sessions for both groups (Figure 3). Small magnitudes of differences were found between the average of intensities (*g* = 0.20) between CG and EG.

## 4. Discussion

The aim of the study was to investigate the effects of a 10-week NMT program on skinfold and body composition variables in highly trained female soccer players. We hypothesized that GE would reduce skinfold values, fat mass, and body mass and increase muscle mass and lean body mass, improving overall body composition.

The main findings of the current work were that 10 weeks of NMT significantly reduced body mass (−0.34%, *g* = −0.04), fat mass (−1.94%, *g* = −0.10), and Σ6S (−1.79%, *g* = −0.09) compared with the CG (0.05%, *g* = 0.01, 0.52%, *g* = 0.02 and 0.4%, *g* = 0.01, respectively). EG and CG were exercised equally, and no significant work intensity was observed between the groups. In addition, body skeletal muscle mass and lean body mass increased in the EG (body skeletal muscle mass: 1.10%, *g* = 0.23; lean body mass: 1.53%, *g* = 0.45) and slightly decreased in the CG (body skeletal muscle mass: −0.10%, *p* = 0.3, lean body mass: −0.10%, *g* = 0.02, respectively).

Previous research in soccer players reported changes in body composition after different resistance training programs [38,40,51,52,53,54,55,56,57,58,59]. However, no study has been conducted regarding the effects of an NMT program. Of note, the NMT battery applied in the current work includes exercises from six categories: (1) mobility, (2) dynamic stability, (3) anterior chain strength, (4) lumbopelvic control, (5) posterior chain strength, and (6) the ability to COD in this regard, and the effectiveness of each or a combination of these training methods to improve body composition has been considered as a reference for comparing the results of the present study.

Arguably, strength exercises are an effective way to stimulate muscle hypertrophy along with improvements in body composition [60]. Particularly, Falces et al. [55] applied a 16-week strength training program with calisthenics and observed a significant decrease in body mass (ES = −0.08) and fat mass (ES = −0.41) and a significant increase in lean mass (ES = 0.17) in a group of male U17 soccer players. Furthermore, Sánchez-Pérez et al. [54] studied the effects of an 8-week high intensity interval training (i.e., a Tabata workout including calisthenics, plyometrics, and COD ability) in a similar population, showing a reduction in body fat (−1.38%, ES = 0.42) and an increase in lean body mass (1.38%, ES = 0.44), and Suárez-Arrones [40,56] also found differences in the body composition (body fat: ES = −0.99 ± 0.54 and lean body mass percentage: ES = 0.25 ± 0.10) of young male soccer players during a 24-week intervention that included circuit training with some exercises comparable to ours (i.e., posterior chain eccentrics, core stability, and plyometrics). It should be noted that in these last two studies the CG slightly worsened their body composition, just as in the present research.

On the contrary, several studies [58,59,61] assessed training programs that include at least one of the exercise categories applied in the current study in adult soccer players showing no differences in changes in body fat percentage (ES = −0.10) and fat-free mass percentage (ES = 0.09) after 8 weeks or less of intervention. Unfortunately, female soccer players were not included in these works, preventing an accurate comparison with the current data.

Focusing on female soccer players, the study from Polman [52] analyzed the effect of a 12-week physical conditioning program on physical fitness and anthropometric parameters of adult highly trained female soccer players. After the intervention, decreases in body mass (ES = −0.24), BMI (ES = −0.28) and fat mass (ES = −0.16) were found. Although the exercise program in Polman’s study is similar to the one included in our study (e.g., balance, jumps, and COD ability), their athletes showed greater improvements than the athletes in the present study. A possible explanation for this little discrepancy could be the longer duration of their intervention and/or the higher body fat percentage of their players at baseline. Remarkably, the mean values for body mass and fat percentage at baseline in the current work fall within the values reported in a review of international female soccer players (56.8–64.9 kg and 14.6–20.1%, respectively) [5], whereas those from the aforementioned study do not.

In contrast, to the best of our knowledge, this is one of the first studies to assess the effects of 12-week plyometric training on body composition, explosive strength, and kicking speed of 20 female soccer players [53]. The results showed an improvement in performance variables but no significant changes in body composition However, changes in muscle strength through plyometric training produce adaptations of the neuromuscular system rather than muscle hypertrophy [62]. Therefore, with unique plyometric training, body composition can be expected to remain unchanged.

Of note, one study [38] analyzed the effects of a 12-week NMT program on the body composition of female volleyball athletes. Though the sports have different metabolic requirements, (football and volleyball), Simões et al. showed an increase in body mass (ES = 0.08) and lean body mass (ES = 0.36) and a reduction in fat mass (ES = −0.50) [38])**.** In the same direction, the study by Sudha and Dharuman, which evaluated the effects of a 12-week circuit training program combined with different neuromuscular activities in schoolgirls, observed a decrease in BMI (ES = −0.49) [51]. This data, although from a different sample, contribute to reinforcing the results obtained in the present research and highlight that the assessment of body composition is closely related to performance and helps to confirm the training effect [62,63].

Some limitations need to be acknowledged for a correct interpretation of the results. Firstly, it should be mentioned that the sample used is small and the data is limited to a certain group of soccer players, so it would be interesting to carry out further studies to confirm the present results. Female soccer players have characteristics that do not allow us to extrapolate our results directly to other sports. This study did not take into account variables related to the genotype of the female athletes and protein intake above the recommended dietary allowance was not controlled. We recommend that future research examines the relationship between different endocrine parameters (i.e., IGFBP-3, erythropoietin, or estrogen for female athletes) and genes related to performance and body composition, such as angiotensin-1 converting enzyme insertion/deletion (ACE I/D) polymorphism or α-actinin-3 (ACTN3) R577X polymorphism. Future studies should extend these observations to other age groups, competitive levels, and larger samples in order to analyze whether the results are similar. Furthermore, it would be beneficial to observe different intensities and volumes in the NMT program to determine the optimal regimen for this training method as well as observing whether this program can improve body composition in female soccer players.

## 5. Conclusions

The present study suggests that the implementation of a 10-week NMT program of just 24 min, three times a week improves body composition in highly trained female soccer players compared to a regular physical preparation training. In this regard, the soccer-specific NMT protocol proposed in this study improved female soccer players´ body composition by reducing fat mass and increasing muscle mass. Therefore, female soccer coaches and physical trainers should be aware that combining strength, mobility, lumbopelvic control, dynamic stability, and change of direction exercises based on soccer-specific requirements may also improve the body composition of their female players.

## Figures and Tables

**Figure 1 biology-11-01062-f001:**
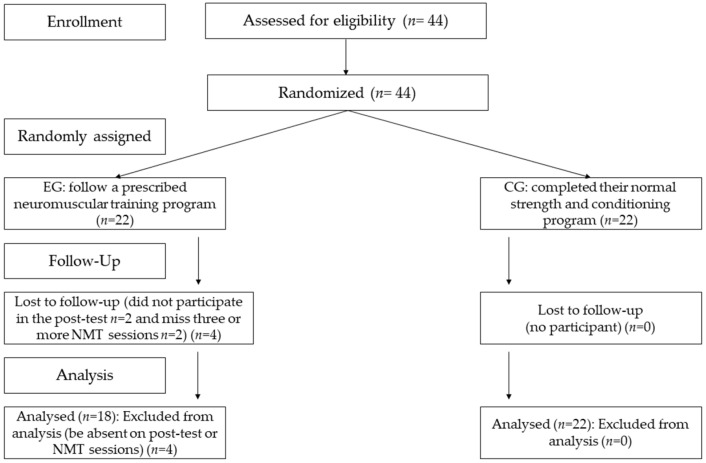
Participant recruitment, allocation, follow-up, and analysis are depicted in a CONSORT diagram. NMT: neuromuscular training; EG: experimental group; CG: control group.

**Figure 2 biology-11-01062-f002:**
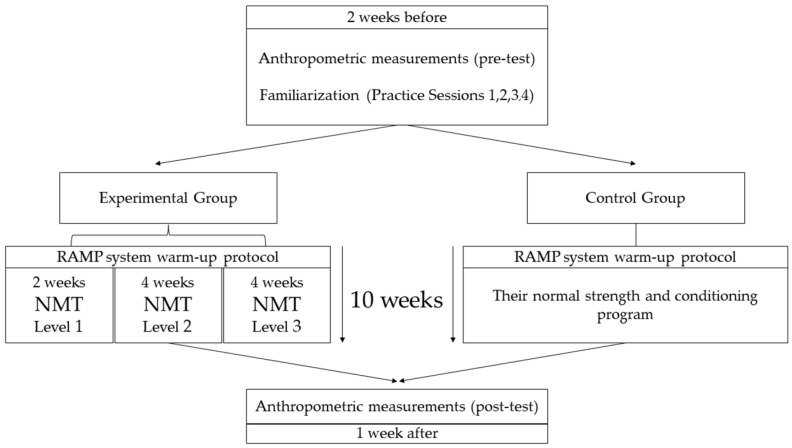
Project design timeline. NMT: neuromuscular training; RAMP: rise, activate, mobilize, and potentiate.

**Figure 3 biology-11-01062-f003:**
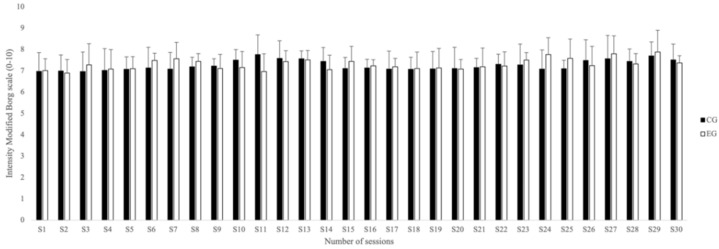
Average intensity (mean and standard deviation) using the modified Borg scale over the thirty training sessions. EG: experimental group; CG: control group.

**Figure 4 biology-11-01062-f004:**
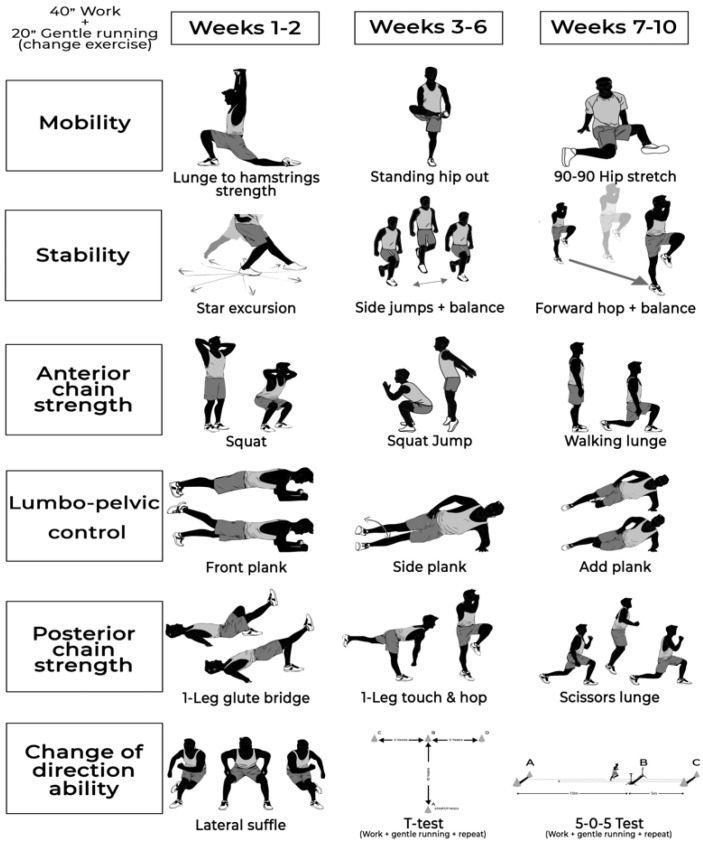
Neuromuscular training protocol.

**Figure 5 biology-11-01062-f005:**
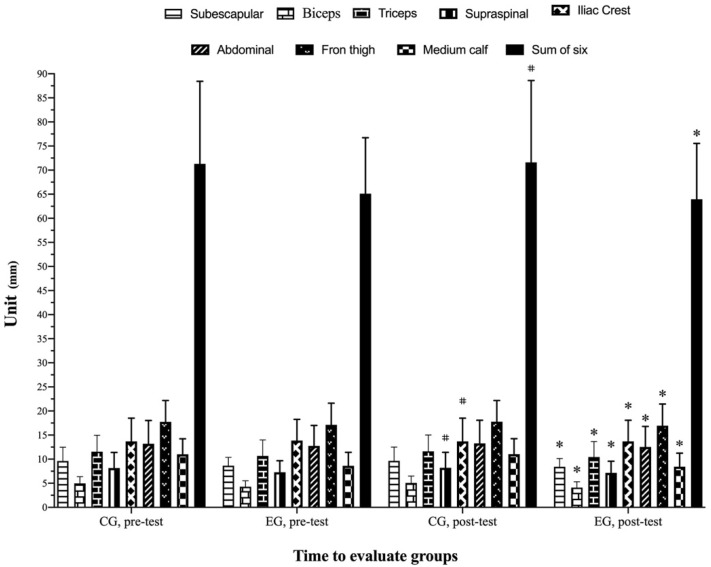
Change in skinfold variables assessment for each group and assessment stage. * Represents a statistically significant difference compared to the pre-test with the superiority of the EG (*p* < 0.05). # Represents a statistically significant difference compared to the pre-test with the superiority of the CG (*p* < 0.05). EG: experimental group; CG: control group.

**Figure 6 biology-11-01062-f006:**
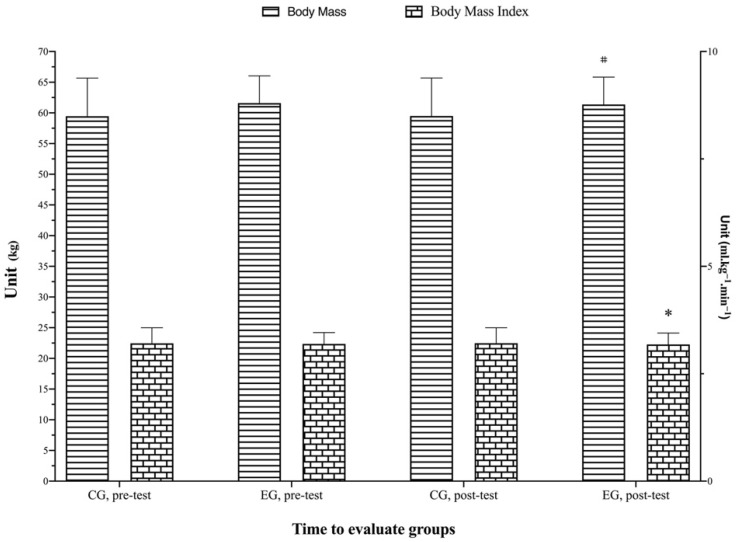
Change in body mass and body mass index variables assessment for each group and assessment stage. * Represents a statistically significant difference compared to the pre-test with the superiority of the EG (*p* < 0.05). # Represents a statistically significant difference compared to the pre-test with the superiority of the CG (*p* < 0.05). EG: experimental group; CG: control group.

**Figure 7 biology-11-01062-f007:**
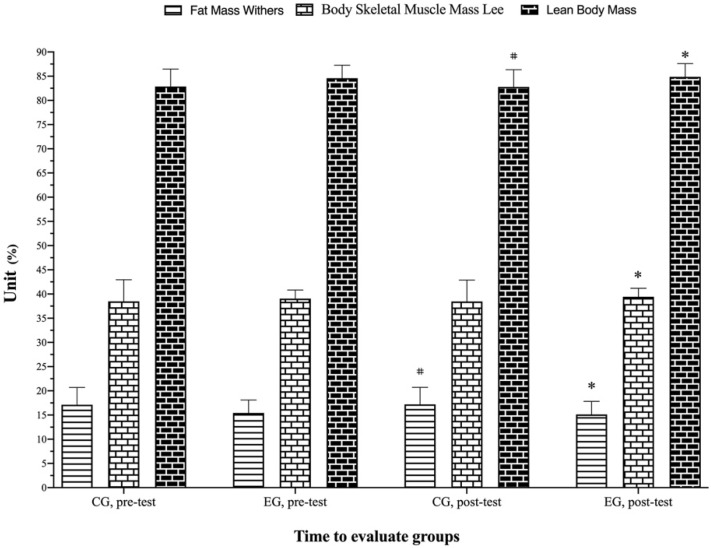
Change in body composition variables assessment for each group and assessment stage. * Represents a statistically significant difference compared to the pre-test with the superiority of the EG (*p* < 0.05). # Represents a statistically significant difference compared to the pre-test with the superiority of the CG (*p* < 0.05). EG: experimental group; CG: control group.

**Table 1 biology-11-01062-t001:** Descriptive data of the participants.

Variable	Control Group (*n* = 22)	Experimental Group (*n* = 18)	*p*
	Mean ± SD	Mean ± SD	
Age (years)	24.61± 4.30	23.24 ± 4	0.31
Height (cm)	162.29 ± 5.90	166.09 ± 4.65	0.07
Body Mass (kg)	59.46 ± 6.22	61.61 ± 4.43	0.23
BMI (kg/m^2^)	22.46 ± 2.54	22.37 ± 1.85	0.89

SD: standard deviation; BMI: body mass index.

**Table 2 biology-11-01062-t002:** Average macronutrient and energy intake.

Variable	Control Group (*n* = 22)		Experimental Group (*n* = 18)		*p*
	1st Registration Mean ± SD	2nd Registration Mean ± SD	1st Registration Mean ± SD	2nd Registration Mean ± SD	
Kilocalories (kcal/day)	2206 ± 377	2222 ± 346	2266 ± 198	2285 ± 189.1	0.48
Carbohydrates (g)	311.9 ± 56.4	315.9 ± 47.9	336.9 ± 29.7	328.8 ± 29.1	0.13
Proteins (g)	92.9 ± 18.2	89.3 ± 13.2	90.8 ± 9.07	90.5 ± 10.6	0.56
Fats (g)	65.1 ± 12.1	66.8 ± 12.8	66.1 ± 6.79	67.8 ± 7.02	0.39

SD: standard deviation; kcal: kilocalories; g: grams.

**Table 3 biology-11-01062-t003:** Training intervention details.

Group	Experimental (*n* = 18)	Control (*n* = 22)			Sum
Training Program	NMT (6 Exercises)	Mobility (3 Exercises)	Strength (3 Exercises)	RT (3 Exercises)	9 Exercises
Training Details	Sets: 4 Work: 40 s Rest: 20 s Duration: 24 min	Sets: 2 Work: 30 s Rest: 20 s Duration: 5 min	Sets: 4 Work: 40 s Rest: 20 s Duration: 12 min	Sets: 3 (4 reps) Work: ~10 s Rest: ~20 s Duration: 7 min	Sets: 2–4 Work: 10–40 s Rest: 20 s Duration: 24 min
Work Intensity (RPE)	7.3 ± 0.25	7.26 ± 0.23			*p* = 0.45

NMT: neuromuscular training; reps: repetitions per set; s: seconds; RPE: rate of perceived exertion (0–10); RT: running technique; SD: standard deviation.

**Table 4 biology-11-01062-t004:** Summary results of skinfold variables within the control group and neuromuscular training group.

Skinfolds (mm)		Control Group (*n* = 22)				Experimental Group (*n* = 18)				
	Pre-Test Mean ± SD	Post-Test Mean ± SD	Pre-Post (%)	*p*	ES (95% CI)	Pre-Test Mean ± SD	Post-Test Mean ± SD	Pre-Post (%)	*p*	ES (95% CI)
Subscapular	9.62 ± 286	9.66 ± 2.85	0.41	0.143	0.01 (−0.64; 0.66) T	8.64 ± 1.71	8.42± 1.73	−2.54	≤0.001 *	−0.12 (−0.77; 0,53) T
Biceps	4.97 ± 1.40	5.07 ± 1.41	2.01	0.008 *	0.06 (−0.58; 0,72) T	4.27 ± 1.23	4.10 ± 1.21	−3.98	0.001 *	−0.13 (−0.78; 0.52) T
Triceps	11.55 ± 3.42	11.63 ± 3.39	0,69	0.144	0.02 (−0.63; 0.67) T	10.67 ± 3.34	10.45 ± 3.21	−2.06	0.018 *	−0.06 (−0.71; 0.59) T
Supraspinal	8.15 ± 3.24	8.22 ± 3.21	0.85	0.046	0.01 (−0.63; 0.67) T	7.29 ± 2.38	7.16 ± 2.40	−1.78	≤0.001 *	−0.05 (−0.70; 0.60) T
Iliac crest	13.61 ± 4.84	13.69 ± 4.84	0.58	0.002 *	0.02 (−0.64; 0.67) T	13.86 ± 4.39	13.68 ± 4.41	−1.29	≤0.001 *	−0.04 (−0.69; 0.61) T
Abdominal	13.21 ± 4.84	13.27 ± 4.81	0.45	0.110	0.01 (−0.66; 0.67) T	12.77 ± 4.23	12.54 ± 4.25	−1.80	0.001 *	−0.06 (−0.70; 0.60) T
Front thigh	17.76 ± 4.42	17.77 ± 4.42	0.05	0.181	0.01 (−0.65; 0.65) T	17.14 ± 4.48	16.95 ± 4.49	−1.10	≤0.001 *	−0.09 (−0.69; 0.61) T
Medial calf	11.03 ± 3.21	11.05 ± 3.20	0.18	0.137	0.01 (−0.65; 0.65) T	8.61 ± 2.82	8.43 ± 2.82	−2.09	≤0.001 *	−0.06 (−0.71; 0,59) T
Σ6S	71.32 ± 17.14	71.61 ± 16.99	0.40	0.019 *	0.01 (−0.64; 0.67) T	65.12 ± 11.61	63.95 ± 11.60	−1.79	≤0.001 *	−0.09 (0.74; 0.55) T

SD: standard deviation; ES: effect size; CI: confidence interval; T: trivial; Σ6S: sum of six skindfolds; * *p* < 0.05.

**Table 5 biology-11-01062-t005:** Summary results of other body composition variables within the control group and neuromuscular training group.

Variable		Control Group (*n* = 22)					Experimental Group (*n* = 18)			
	Pre-Test Mean ± SD	Post-Test Mean ± SD	Pre-Post (%)	*p*	ES (95% CI^4^)	Pre-Test Mean ± SD	Post-Test Mean ± SD	Pre-Post (%)	*p*	ES (95% CI)
Body mass (kg)	59.46 ± 6.22	59.49 ± 6.21	0.05	0.468	0.01 (−0.65; 0.65) T	61.59 ± 4.44	61.37 ± 4.45	−0.35	≤0.001 *	−0.04 (−0.70; 0.60) T
BMI (kg/m^2^)	22.46 ± 2.54	22.47 ± 2.53	0.04	0.497	0.01 (−0.65; 0.66) T	22.36 ± 1.85	22.28 ± 1.81	−0.35	≤0.001 *	−0.04 (−0.69; 0,61) T
Fat mass Withers (%)	17.13 ± 3.57	17.21 ± 3.53	0.52	0.029 *	0.02 (−0.64; 0.68) T	15.42 ± 2.68	15.12 ± 2.71	−1.94	≤0.001 *	−0.10 (−0.75; 0.54) T
Body skeletal muscle mass Lee (%)	38.50 ± 4.47	38.45 ± 4.43	−0.10	0.309	0.01 (−0.66; 0.64) T	39.03 ± 1.78	39.46 ±1.75	1.10	≤0.001 *	0.23 (−0.42; 0,88) S
Lean body mass (%)	82.87 ± 3.57	82.79 ± 3.53	−0.10	0.029 *	−0.02 (−0,67; 0.63) T	84.58 ± 2.68	85.88 ± 2.71	1.53	<0.001 *	0.45 (−0.20; 1,12) S

SD: standard deviation; BMI: body mass index; ES: effect size; CI: confidence interval; T: trivial; S: small * *p* < 0.05.

## Data Availability

The data presented in this study are available on reasonable request from the corresponding author. The data are not publicly available due to privacy reasons.

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
