# Peer review of "Evaluation of 10-Week Neuromuscular Training Program on Body Composition of Elite Female Soccer Players"

_biology, 2022, doi:10.3390/biology11071062_

Round 1

Reviewer 1 Report

The authors or this manuscript has conducted that evaluation of 10-week neuromuscular training program on body composition of elite female soccer players. Due to the difficulties in establishing the evaluation strategy for exercise training, this is an interesting topic through neuromuscular training. However, this manuscript still has problems for experimental design and data analysis, as shown in following points:

1.       The shortcoming of this manuscript does not fully concentrate on the focus of key scientific questions of evaluating the training strategy, such as specific parameters including genes or proteins for exercise performance related to neuromuscular signaling during training process, rather than the simple body compositions alone, which may greatly reduce the creativity of this manuscript.

2.       The data analysis revealed the very significant difference with P value less than 0.001 during the statistical analysis of data. I don’t believe that it can have very significant difference and I don’t know which statistical methods were used. For example, between 13.86 ± 4.39 and 13.68 ± 4.41, between 7.29 ± 2.38 and 7.16 ± 2.40, between 17.14 ± 4.48 and 16.95 ± 4.49, between 8.64 ± 1.71 and 8.42± 1.73 in Table 3, and so on. Therefore, I doubt its training effect with very significant difference based on statistical analysis.

Author Response

Dear Reviewer,

We have carefully considered all considerations in the document provided by you. Enclosed you will find our detailed answers to your inquiries.

Thank you for the time taken to review our paper and for giving us the chance to improve it.

We respond point by point below.

Comments and suggestions for Authors

The authors or this manuscript has conducted that evaluation of 10-week neuromuscular training program on body composition of elite female soccer players. Due to the difficulties in establishing the evaluation strategy for exercise training, this is an interesting topic through neuromuscular training. However, this manuscript still has problems for experimental design and data analysis, as shown in following points:

  1. The shortcoming of this manuscript does not fully concentrate on the focus of key scientific questions of evaluating the training strategy, such as specific parameters including genes or proteins for exercise performance related to neuromuscular signaling during training process, rather than the simple body compositions alone, which may greatly reduce the creativity of this manuscript.

Authors: Thank you for your comment. We have included a new sentence in the introduction.  Also, we have added it as limitations, " This study did not take into account variables related to the genotype of the female athletes and protein intake above the recommended dietary allowance was not controlled."

  1. The data analysis revealed the very significant difference with P value less than 0.001 during the statistical analysis of data. I don’t believe that it can have very significant difference and I don’t know which statistical methods were used. For example, between 13.86 ± 4.39 and 13.68 ± 4.41, between 7.29 ± 2.38 and 7.16 ± 2.40, between 17.14 ± 4.48 and 16.95 ± 4.49, between 8.64 ± 1.71 and 8.42± 1.73 in Table 3, and so on. Therefore, I doubt its training effect with very significant difference based on statistical analysis.:

Authors: Thank you for your comment. For your information, in the follow SPSS tables you can observe the mean, standard deviation, and p-value of each variable after student paired t-test in the experimental group.

The authors hope that the corrections made are to your satisfaction and we can continue with the peer-review process of the manuscript.

Best regards.

Reviewer 2 Report

Introduction:

-       Define what a license is.

-       Not sure the first paragraph is necessary. It leads the reader down a different path than the purpose of the paper

-       Lines 53-55: Sentence should be re-worded as it seems a bit jumbled

-       Line 57: change “These” to “Previous”

-       Line 57: What is meant by blurry link? Provide an example from the referenced studies

-       Line 63: insert comma after ratio

-       Line: 66: elite male, youth, and female what, what kind of athlete, be specific. What was the level of competition for the female athletes referenced?

-       Line 68: change “some” to “previously”

-       Lines 72-75: This sentence is too long and could be broken up to make your point clearer to the reader

-       Line 80: did you mean decade? If not, define how many decades you are talking about.

-       Lines 80-81: This sentence is incomplete- add “have been developed” to the end.

-       Line 83: add a comma after agility

-       Line 88: change “stablished” to “established”

-       Line 90: change “female” to “females”

-       Lines 89-94: These statements feel thrown in the paragraph with no transition to the topics

-       Line 97: take “maybe” out of the sentence

Materials and Methods:

-       Lines 104-105: Do not need the characteristics here as they are in Table 1

-       Lines 110-111: Re-word this sentence, it is confusing.

-       Line 121: add “to” between “prior” and “data”

-       Lines 173-174: Remove “again” from the sentence

-       Change “student” to “test” in the statistical analysis section

-       How did you collect macronutrient and energy intake? This is not in your methods.

-       This section appears to have been written by a different person. While collaborative work is necessary, the same style of writing should be carried throughout the paper.

Include the specific exercises and exercise volume of the regular physical preparation program.

Results:

-       Put the results for each variable after the variable in their own parentheses. It is too much to look at when they are clumped into one. This will more clearly show your results to the reader.

Discussion:

-       Lines 295-296: what kind to training programs?

-       Line 296: change “found” to “conducted”

-       Line 297: What do you mean “making this section particularly challenging”? If you are referring to writing the discussion, remove this from the sentence.

-       Lines 300-302: This should be referenced and you should know if they have been previously evaluated from conducting a review of literature prior to completing this study.

-       Line 317: change “a few” to “several”

-       Line 328: change “their results were higher than ours” to “their athletes showed greater improvements than the athletes in the present study”

-       Line 329: remove “little”

-       Line 329: change “and” to “and/or”

-       Line 334: change “the” after to the best of to “our”

-       Line 334: change “study” to “studies”

-       Lines 337-340: sentence is too long and should be re-worded

-       Lines 342-343: change “even though the sports are not fully comparable” to “Though the sports have different metabolic requirement”

-       Line 343: remove “this work from” from the sentence

-       Lines 342-347: Entire sentence needs restructuring. It seems like the first thought was not finished and another one began within the same sentence

-       Lines 351-360: You did not really state limitations to your study, but rather stated where future research should focus. This is not appropriate for this paragraph

Author Response

Dear Reviewer,

We have carefully considered all considerations in the document provided by you. Enclosed you will find our detailed answers to your inquiries.

Thank you for the time taken to review our paper and for giving us the chance to improve it.

We respond point by point below.

Comments and suggestions for Authors

Introduction:

-       Define what a license is

Authors: Thanks for your comment. A sport license is the document from the National Federation that allowed you to play in the national and international competitions. We have changed the term “license” for “federated players”.

-       Not sure the first paragraph is necessary. It leads the reader down a different path than the purpose of the paper

Authors: The authors agree. We have rewritten this paragraph.

-       Lines 53-55: Sentence should be re-worded as it seems a bit jumbled

Authors: You are right. The paragraph now is as follows: “Of note, these high-intensity actions, coupled with the ability to repeat them without fatigue, also account for most of the injury inciting events [7]. In addition, the somatotype of the players has shown to be relevant for this purpose [8]”

-       Line 57: change “These” to “Previous”

Authors: Changed.

-    Line 57: What is meant by blurry link? Provide an example from the referenced studies

Authors: Thanks. The actual term used was “burly link”, trying to highlight the strong connection between both events. It has been changed for “strong” for increasing the clarity of the sentence.

-       Line 63: insert comma after ratio.

Authors: Changed.

-       Line: 66: elite male, youth, and female what, what kind of athlete, be specific. What was the level of competition for the female athletes referenced?

Authors: Thanks for your comment. This information has been added in the text.

-       Line 68: change “some” to “previously”

Authors: Changed.

-       Lines 72-75: This sentence is too long and could be broken up to make your point clearer to the reader

Authors: You are right. The paragraph now is as follows: “The research working group on body composition health and performance of athletes state that low body fat and high lean body mass are strongly correlated with higher levels of performance, especially in weight-sensitive sports such as soccer [23]”

-       Line 80: did you mean decade? If not, define how many decades you are talking about.

Authors: Changed to "In the last two decades".

-       Lines 80-81: This sentence is incomplete- add “have been developed” to the end

Authors: Added.

-       Line 83: add a comma after agility.

Authors: Added.

-       Line 88: change “stablished” to “established”

Authors: Changed.

-       Line 90: change “female” to “females”

Authors: Changed.

-       Lines 89-94: These statements feel thrown in the paragraph with no transition to the topics

Authors: You are right. The paragraph now is as follows: “Rohmansyah et al. [36] found a reduction in body mass index (BMI), fat mass, and waist circumference in obese young-adult females after a 6-week FIFA 11+ program. Simões et al. analyzed the effects of NMT on body composition in volleyball athletes finding improvements in body composition [37]. With respect to each of the interventions included in a NMT program, there is some research on the effects of body-weight resistance training [38], eccentric [39], and plyometric based programs [40] on the body composition of female soccer players. However, the evidence is very scarce regarding the effects of a multicomponent program that combines all of them”

-       Line 97: take “maybe” out of the sentence.

Authors: Removed.

Materials and Methods:

-       Lines 104-105: Do not need the characteristics here as they are in Table 1.

Authors: Removed.

-       Lines 110-111: Re-word this sentence, it is confusing.

Authors: Thank you. It has been clarified.

-       Line 121: add “to” between “prior” and “data”

Authors: Added.

-       Lines 173-174: Remove “again” from the sentence.

Authors: Removed.

-       Change “student” to “test” in the statistical analysis section

Authors: Thanks. Changed.

-       How did you collect macronutrient and energy intake? This is not in your methods.

Authors: Thank you for your comment. We have added this information in the method section.

-       This section appears to have been written by a different person. While collaborative work is necessary, the same style of writing should be carried throughout the paper.

Authors: Thank you for your comment. We have reviewed this section.

Include the specific exercises and exercise volume of the regular physical preparation program.

Authors: Thank you. We added it.

Results:

-       Put the results for each variable after the variable in their own parentheses. It is too much to look at when they are clumped into one. This will more clearly show your results to the reader.

Authors: Thank you for your comment. We follow your suggestion and also, another reviewer suggestion that he/she asked to reduce the result section.

Discussion:

-       Lines 295-296: what kind to training programs?

Authors: Thank you. It has been clarified.

-       Line 296: change “found” to “conducted”

Authors: Changed.

-       Line 297: What do you mean “making this section particularly challenging”? If you are referring to writing the discussion, remove this from the sentence.

Authors: Removed.

-       Lines 300-302: This should be referenced and you should know if they have been previously evaluated from conducting a review of literature prior to completing this study.

Authors: Thanks for your comment. The sentence has been updated to clearly state that these are the articles that will be considered to compare their results with ours.

-       Line 317: change “a few” to “several”

Authors: Changed.

-       Line 328: change “their results were higher than ours” to “their athletes showed greater improvements than the athletes in the present study”

Authors: Changed.

-       Line 329: remove “little”.

Authors: Removed.

-       Line 329: change “and” to “and/or”

Authors: Changed.

-       Line 334: change “the” after to the best of to “our”

Authors: Changed.

-       Line 334: change “study” to “studies”

Authors: Changed.

-       Lines 337-340: sentence is too long and should be re-worded

Authors: Rephrased the sentence

-       Lines 342-343: change “even though the sports are not fully comparable” to “Though the sports have different metabolic requirement”

Authors: Changed.

-       Line 343: remove “this work from” from the sentence.

Authors: Removed.

-       Lines 342-347: Entire sentence needs restructuring. It seems like the first thought was not finished and another one began within the same sentence

Authors: Restructured

-       Lines 351-360: You did not really state limitations to your study, but rather stated where future research should focus. This is not appropriate for this paragraph

Authors: Thanks to your insightful revision and comments a few limitations have been added to this paragraph to make it more suitable.

The authors hope that the corrections made are to your satisfaction and we can continue with the peer-review process of the manuscript.

Best regards.

Reviewer 3 Report

An interesting piece of work that addresses a practically valid research question. I like the use of control groups in order to offer a meaningful comparison of the effects of the intervention. I think this study has value and would be appreciated by practitioners and researchers in the field. That said I would like to suggest some changes.

1.      In the introduction they suggest that reduced body fat is associated with better sports performance indicating soccer as an example. I agree with the sentiment that it can improve fitness in terms of speed and strength.  However, I suggest some caution in the assumption that there is a linear relationship between percentage body fat and performance. Individuals with a higher percentage body fat can improve their performance by reducing it whereas people who already have relatively low percentage body fat may not. In fact, among people with a low body fat, trying to lose weight might be detrimental. I suggest re-visiting this aspect of the introduction.

2.      I would like to see the authors explain the mechanism through which the intervention would lead to improved performance and from there I would like a stronger justification an explanation on how the intervention was put together. I do not disagree with the intervention presented but would want to know which of these exercises activate the essential ingredients in order to bring about positive change. I suspect we do not know the answer to this question, and as such this could be raised as a limitation an area of future investigation.

3.      I found the statistics heavy going to get through and would suggest that the authors look to remove as much reporting of statistical results in the text and move this information into tables. I suggest they depict the results graphically so that the reader could follow all the results are saying more easily. At the present time the reader has to work quite hard in order to find out the results are saying.

4.      It might help the authors if a specific hypothesis can be set around each statistical test. And as it is a multivariate question, therefore is it possible to use a repeated measures man over. 7

5.      Please provide more information on what it means to be a semi-professional.

Author Response

Dear Reviewer,

 We have carefully considered all considerations in the document provided by you. Enclosed you will find our detailed answers to your inquiries.

Thank you for the time taken to review our paper and for giving us the chance to improve it.

We respond point by point below.

Comments and suggestions for Authors

An interesting piece of work that addresses a practically valid research question. I like the use of control groups in order to offer a meaningful comparison of the effects of the intervention. I think this study has value and would be appreciated by practitioners and researchers in the field. That said I would like to suggest some changes.

  1. In the introduction they suggest that reduced body fat is associated with better sports performance indicating soccer as an example. I agree with the sentiment that it can improve fitness in terms of speed and strength.However, I suggest some caution in the assumption that there is a linear relationship between percentage body fat and performance. Individuals with a higher percentage body fat can improve their performance by reducing it whereas people who already have relatively low percentage body fat may not. In fact, among people with a low body fat, trying to lose weight might be detrimental. I suggest re-visiting this aspect of the introduction.

Authors: Thank you for your comment. The paragraph now is as follows: “The research working group on body composition health and performance of athletes state that low body fat and high lean body mass are strongly correlated with higher levels of performance, especially in weight-sensitive sports such as soccer [23]. However, this relationship should be taken cautiously as every sport has its own body composition (i.e., somatotype) that is considered ideal for success [24], and players with low body fat mass whould not follow this general rule. In this regard, a recently published narrative review included kinanthropometric data of elite female soccer players from 2000 to 2020, show-ing a fat mass percentage between 14.5% and 22% [25]”.

  1. I would like to see the authors explain the mechanism through which the intervention would lead to improved performance and from there I would like a stronger justification an explanation on how the intervention was put together. I do not disagree with the intervention presented but would want to know which of these exercises activate the essential ingredients in order to bring about positive change. I suspect we do not know the answer to this question, and as such this could be raised as a limitation an area of future investigation.

Authors: Thanks for your comment. The rationale of applying neuromuscular training programs in team sports are twofold: to improve sport-specific performance and to reduce injury risk. It is already explained in the text (lines 79-82), but the authors agree with the reviewer´s suspicion about which of these exercises are the most important to bring a positive change (definitely not the mobility exercises), so a comment has been added in the limitations section.

  1. I found the statistics heavy going to get through and would suggest that the authors look to remove as much reporting of statistical results in the text and move this information into tables. I suggest they depict the results graphically so that the reader could follow all the results are saying more easily. At the present time the reader has to work quite hard in order to find out the results are saying.

Authors: Thank you for your comment. We have improved this section.

  1. It might help the authors if a specific hypothesis can be set around each statistical test. And as it is a multivariate question, therefore is it possible to use a repeated measures man over.

Authors: Thank you for your comment. We added some information about the hypothesis. In the terms of the analysis we have examined a 2 (group) x 2 (time) repeated measures ANOVA.

  1. Please provide more information on what it means to be a semi-professional.

Authors: Thanks for this comment. It is our mistake. Semi-professional has been removed from the test and substituted by “highly-trained” following the algorithm from Mckay et al 2022 (McKay, A. K., Stellingwerff, T., Smith, E. S., Martin, D. T., Mujika, I., Goosey-Tolfrey, V. L., ... & Burke, L. M. (2022). Defining Training and Performance Caliber: A Participant Classification Framework. International journal of sports physiology and performance, 17(2), 317-331).

The authors hope that the corrections made are to your satisfaction and we can continue with the peer-review process of the manuscript.

Best regards.

Round 2

Reviewer 1 Report

Since I have made the rejection due to its creativity and poor data presentation as well as unreliable statistical analysis, I do not believe the revised manuscript has conducted the substantial modifications to address my concerns. Therefore, I still think this manuscript is not suitable for being accepted. 

Author Response

Dear Reviewer,

We have carefully considered all considerations in the document provided by you. Enclosed you will find our detailed answers to your inquiries.

Thank you for the time taken to review our paper and for giving us the chance to improve it.

Reviewer 2 Report

This paper is much improved. 

Please consider adding a table with the exact volume (set and reps) of the control group exercises. The NMT may have results in improvements simply because the athletes did more training. This point needs to be clarified in the Results and Discussion.

Also, the grammar and spelling still needs to be improved. 

Author Response

Dear Reviewer,

We have carefully considered all considerations in the document provided by you. Enclosed you will find our detailed answers to your inquiries.

Thank you for the time taken to review our paper and for giving us the chance to improve it.

We respond point by point below.

Comments and suggestions for Authors

Comment 1:    Please consider adding a table with the exact volume (set and reps) of the control group exercises. The NMT may have results in improvements simply because the athletes did more training. This point needs to be clarified in the Results and Discussion.

Authors: Thanks for your comment. We added table 3. In addition, we monitored and control training load to avoid of difference load between groups. You can see in figure 3.

Comment 2: The grammar and spelling still needs to be improved. 

Authors: Thanks for your comment. The English manuscript has been reviewed by a native person.

Round 3

Reviewer 2 Report

Thank you for the revisions. Nice job.

Author Response

(The authors gave the same response as above.)
